# DIC-Like Syndrome Following Administration of ChAdOx1 nCov-19 Vaccination

**DOI:** 10.3390/v13061046

**Published:** 2021-06-01

**Authors:** Gerardo Casucci, Domenico Acanfora

**Affiliations:** Unit of Internal Medicine, San Francesco Hospital, Viale Europa 21, 82037 Telese Terme, Italy; domenico.acanfora29@gmail.com

**Keywords:** SARS-CoV-2, chimpanzee adenovirus vectored vaccine, disseminated intravascular coagulation

## Abstract

In recent weeks, adverse reactions have been reported after administration of Oxford–AstraZeneca chimpanzee adenovirus vectored vaccine ChAdOx1 nCoV-19 (AZD1222), in particular thrombus formation, which has led several European Countries to discontinue administration of this vaccine. On March 8, 2021, the European Medicines Agency Safety Committee did not confirm this probable association. We report the case of a patient who developed disseminated intravascular coagulation after the first dose of Oxford-Astra Zeneca vaccine, which resolved in a few days with the administration of dexamethasone and enoxaparin. This work demonstrates the safety of the Oxford-Astra Zeneca vaccine and that any development of side effects can be easily managed with a prompt diagnosis and in a short time with a few commonly used drugs.

## 1. Introduction

The Severe Acute Respiratory Syndrome Coronavirus 2 (SARS-CoV2) has spread quickly around the world, causing clusters of prevalent respiratory Coronavirus Disease 2019 (COVID-19), including Acute Respiratory Distress Syndrome (ARDS), and becoming a serious public health concern. The development and subsequent use of anti-COVID-19 vaccines is the winning weapon to slow down and stop the Sars-CoV-2 pandemic.

Currently, 4 anti-COVID-19 vaccines are available in Europe that use different platforms to deliver mRNA SARS-CoV-2 Spike (S) protein (Table 1).

From 7 to 18 March 2021, some European Countries suspended the inoculation of the AstraZeneca anti-SARS-CoV-2 vaccine. This measure came after some serious cases of blood clots were reported in people who were vaccinated with AstraZeneca’s SARS-CoV-2 vaccine. At the moment, a link between the vaccine and blood clots is not ascertained; however, it has been decided to perform further investigations. The first country to do so was Austria which was then joined by several other countries (Bulgaria, Cyprus, Denmark, Estonia, France, Greece, Iceland, Ireland, Latvia, Luxembourg, Malta, Netherlands, Poland, Spain, Sweden, Estonia, Lithuania, Luxembourg and Italy). On March 18, 2021, the European Medicines Agency Safety Committee reported that the benefits of AstraZeneca’s SARS-CoV-2 vaccine far outweigh the risks of side effects and above all the thrombotic risk. The vaccine may be associated with rare cases of thrombocytopenia, with or without bleeding, including rare cases of cerebral venous thrombosis. Various platforms for vaccine development are available namely: virus vectored vaccines, protein subunit vaccines, genetic vaccines, and monoclonal antibodies for passive immunization which are under evaluation for SARS-CoV-2, each having demonstrated discrete benefits and hindrances. Oxford AstraZeneca consists of a replication-deficient chimpanzee adenoviral vector ChAdOx1, containing the SARS-CoV-2 structural surface glycoprotein antigen (spike protein; nCoV-19) gene (AZD1222). A replication-deficient chimpanzee adenoviral vector ChAdOx1 has already been used for the development of several vaccines [1,2,3,4,5,6,7,8]. Feng-Cai Zhu et al. report that the occurrence of side effects in subjects inoculated with Ad5-vectored SARS-CoV-2 vaccine was associated with decreasing age and low pre-existing immunity to the vaccine vector Ad5virus [9]. Phase 1/2 and 3 studies report very rare cases of arterial or venous thrombosis after administration of Oxford AstraZeneca chimpanzee adenovirus vectored vaccine ChAdOx1 nCoV-19 (AZD1222) [10]. We report the case of a 52-year-old woman with disseminated intravascular coagulation (DIC) [10,11] after administration of the first dose of AstraZeneca anti-SARS-CoV-2 vaccine (Lot ABV4678) in the absence of thrombus in the circulatory system.

## 2. Case History

A 52-year-old woman with a medical history of hepatitis B, headache, left breast cancer treated with bilateral mastectomy, left ovarian cyst treated with oophorectomy and salpingectomy, no prescription of chemotherapy or estrogen–progestogen treatment. The patient remained asymptomatic for the past five years and without any noteworthy diseases; among other things, she was a blood donor. On February 22, 2021, she received the first dose of AstraZeneca anti-SARS-CoV-2 vaccine (Lot ABV4678). Within hours of the administration of the vaccine, the patient presented with throbbing headache, photophobia, nausea, chills, fever (39 °C), muscle and joint pain, and inability to walk mainly due to severe asthenia. The patient was treated at home with paracetamol, ibuprofen and ketoralac. After 48 h, the fever regressed with persistence of all other symptoms (especially an intractable headache). Due to the worsening of symptoms and the appearance of a large ecchymosis of the left buttock on March 9, 2021, the patient was hospitalized.

On admission, vital signs were as follows: blood pressure, 130/80 mm Hg; heart rate, 64 beats/min; respiratory rate, 18/min, and temperature 36.8 °C. The patient tested negative for SARS-Cov-2 on molecular swab.

No abnormalities were detected on echocardiography, arterial and venous ultrasound Doppler in the lower limbs, abdominal ultrasound, ultrasound Doppler of the supra-aortic vessels, and CT scan of the chest and brain. At MRI angiography of the intra and extracranial vessels, no relevant abnormalities were found, while in the venous phase only a hypoplasia of the left transverse sinus was found, in the absence of thrombus (Figure 1).

Based on laboratory data, the disseminated intravascular coagulation score (DIC Score = 5) was calculated [12], suggestive of overt DIC.

On March 9, 2021, the patient was started on subcutaneous enoxaparin (55 mg/day) and intravenous dexamethasone (8 mg/day).

Table 2 summarizes laboratory data before and during the disease. After six days of treatment, all laboratory parameters associated with DIC returned to normal and the patient was asymptomatic. On the seventh day, she was discharged without any home therapy. At discharge, the total IgG assay directed against the Spike protein RBD receptor was 15.1 U/mL (reference range < 0.80 /mL).

Venous thrombosis, including deep vein thrombosis (DVT), rare cases of cerebral venous thrombosis (CVT) and pulmonary embolism (PE), occur at an annual incidence of about 1 per 1000 adults [13]. After administration of Oxford AstraZeneca chimpanzee adenovirus vectored vaccine ChAdOx1 nCoV-19, only 660 thrombotic events were reported in around 11 million people, with an incidence of 6 per 100,000 vaccinated subjects [14,15]. DIC is an acquired syndrome characterized by disordered blood coagulation. Insults or injuries that can lead to DIC can be infectious or non-infectious disorders. To the best of our knowledge, this is the first report of DIC after administration of the first dose of Oxford AstraZeneca chimpanzee adenovirus vectored vaccine ChAdOx1 nCoV-19.

In the patient observed at our Institute we observed a rapid remission of the symptoms and normalization of the blood chemical parameters, in particular those relating to DIC.

Oxford AstraZeneca chimpanzee adenovirus vectored vaccine ChAdOx1 nCoV-19 (AZD1222) can potentially induce a DIC-like syndrome due to the vector used to transfer the genetic substrate for SARS-CoV-2 immunization [1,2,3,4,5,6,7,8]. The chimpanzee adenovirus may induce a pro-coagulative state and a subsequent DIC due to the lack of previous exposure to the specific adenovirus. Binding of adenovirus to platelets can cause platelet activation and thrombosis [16].

Previous studies [1,2,3,4,5,6,7,8] have shown that the surface antigenic structure of chimpanzee adenovirus could influence the immune response of the vector and induce an abnormal immune reaction to immunization. Chimpanzee adenovirus vaccine platforms offer a good safety profile in phase I in humans, efficient bio-manufacture of millions of doses of non-replicating adenoviral vectors is possible using Good Manufacturing Practice (GMP)-approved cell lines. They avoid pre-existing immunity to human adenoviruses, and a single non-adjuvated dose of ChAdOx1 nCoV-19 vaccine is enough to achieve remarkable breadth, durability and potency of both humoral and cellular immune response.

The DIC in our patient is likely to be associated with an age under 60 and low pre-existing immunity to the vaccine vector Ad5 virus [16].

The Oxford AstraZeneca chimpanzee adenovirus vectored vaccine ChAdOx1 nCoV-19 appears to be effective and safe for SARS-CoV-2 immunization [10] and the development of a DIC can be easily controlled by the administration of dexamethasone and enoxaparin for a short time. The timeliness of diagnosis is therefore the best way to reassure patients. For this purpose, blood levels of fibrinogen and d-dimer and platelet counts may constitute more than reliable biological markers of onset or ongoing DIC.

## Figures and Tables

**Figure 1 viruses-13-01046-f001:**
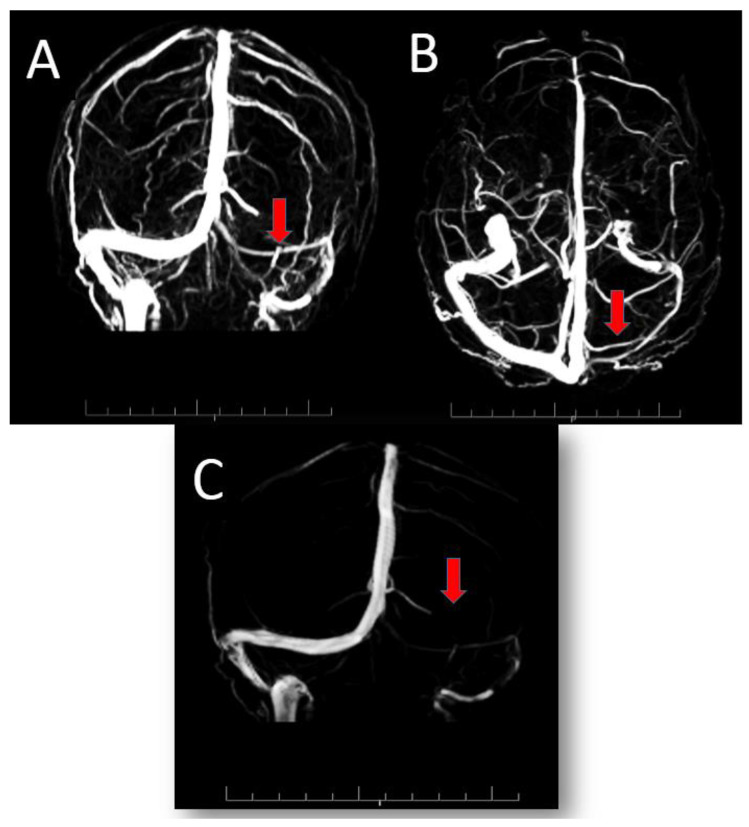
3D reconstruction of venous angio-RM showing hypoplasia of left transverse sinus, indicated in (**A**) (Coronal), (**B**) (Caudocranial) and (**C**) (Sagittal) figures by the red arrows, in the absence of thrombus.

**Table 1 viruses-13-01046-t001:** COVID-19 Vaccines available in Europe.

Manufacturer	Name of Vaccine	Platform
BioNTech/Fosun Pharma/Pfizer	BNT162b2/COMIRNATY (INN tozinameran)	Nucleoside modified mNRA
Moderna/National Institute of Allergy and Infectious Diseases	mRNA-1273	mNRA-based vaccine encapsulated in lipid nanoparticle (LNP)
University of Oxford/AstraZeneca	AZD1222	Recombinant replication defective chimpanzee adenovirus expressing the SARS-CoV-2 S surface glycoprotein
Janssen Pharmaceutical Companies	Ad26.COV2.S	Recombinant, replication incompetent adenovirus type 26 (Ad26) vectored vaccine encoding the (SARS-CoV-2) Spike (S) protein

**Table 2 viruses-13-01046-t002:** Summary of Patient Laboratory Data Related to Disseminated Intravascular Coagulation.

**Laboratory Values (Reference Range)**	**13 December 2020**	**22 February 2021** **Administration of First dose of AstraZeneca anti-SARS-CoV2 vaccine (Lot ABV4678)**	**9 March 2021**	**10 March 2021**	**11 March 2021**	**12 March 2021**	**13 March 2021**	**15 March 2021**	**16 March 2021**
White Blood Cells count (3.7–10.3), ×10^9^/L	6.88		12.52	13.15	10.78	8.11	9.63	10.71	
Red Blood Cells count (4.0–10.0), ×10^6^/L	4.22		4.3	3.80	3.53	3.79	3.69	3.86	
Haemoglobin (13.7–17.5), g/dL	13.3		12.6	11.2	10.4	11.1	10.7	11.4	
Platelet count (155–369), ×10^9^/L	347		77	91	79	108	137	265	330
Prothrombin time (9.6–12.5), s			15.0	15.9				14.1	
International normalized ratio (0.9–1.2)			1.12	1.19				1.05	
Activated partial Thromboplastin time (19–30), s			28.0	28.1				25.5	
Fibrinogen (150–450), mg/dL			100	85	78	90	111	123	248
Lactate dehydrogenase (140–280), U/L			579	587	485	514	461	316	
Creatinine (0.8–1.30), mg/dL	0.8		0.54	0.54	0.53	0.76	0.67		
Aspartate Aminotrasferase (0–31), U/L	26		37	32	27	20	20		
Alanine Aminotrasferase (0–34), U/L	18		43	38	36	28	28		
High Sensitivity C Reactive Protein (0–45), mg/L			2.42	2.05	1.21	0.85	0.62	0.29	
Sodium (135–155), mEq/L			140						
Potassium (3.5–5.5), mEq/L			4.1						
D-dimer (250–500), ng/mL			8298	6481	5280	6187	5128	1624	460
DIC Score			5	6				0	

## Data Availability

Data sharing is not applicable to this case report.

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
