# Peer review of "DIC-Like Syndrome Following Administration of ChAdOx1 nCov-19 Vaccination"

_viruses, 2021, doi:10.3390/v13061046_

Round 1

Reviewer 1 Report

Most of the problems to be concerned are not be responded, and the few responses are also off the point. Taken together, it is not recommended to be pressed.

Line 62: What does the molecular swab mean?

D-dimer in table 1: The values of D-dimer should be translated into ng/ml same as the listed range. In other words, all the punctuation symbols of "." in the numbers should be corrected as ",".

Line 83-85: no the identical or approximative expression "660 thrombotic events in around 11 million people" are found in this mentioned reference 14, please re-confirmed and modified 100.000 as 100,000, too.

Line 96-98: the explanation for the observed DIC is against common sense.

Line 107-108: the association of pre-existing immunity to Ad5 with the DIC isn’t convincing.

Author Response

We thank this Reviewer for the constructive comments and suggestions. Furthermore, we would like to really thank him/her for his/her appreciation about our research in the introduction section of his/her comments. This is our point to point reply.

Line 62: What does the molecular swab mean?

Done.

We added SARS-CoV-2.

D-dimer in table 1: The values of D-dimer should be translated into ng/ml same as the listed range. In other words, all the punctuation symbols of "." in the numbers should be corrected as ",".

Done.

Line 83-85: no the identical or approximative expression "660 thrombotic events in around 11 million people" are found in this mentioned reference 14, please re-confirmed and modified 100.000 as 100,000, too.

Done.

We added reference 15 and changed 100.000 to 100,000.

European Medicines Agency. Covid-19 vaccine AstraZeneca: PRAC investigating cases of thromboembolic events vaccine’s benefits currently still outweigh risks: update. Mar 2021. https://www.ema.europa.eu/en/news/covid-19-vaccine-astrazeneca-prac-investigating-casesthromboembolic-events-vaccines-benefits.

Line 96-98: the explanation for the observed DIC is against common sense.

Done.
We expanded the hypothesis of the immune response induced by Ad5 virus and added the reference.

The chimpanzee adenovirus may induce a procoagulative state and a subsequent DIC due to the lack of previous exposure to the specific adenovirus. Binding of adenovirus to platelets, causing platelet activation and thrombosis [17].

Signal assessment report on embolic and thrombotic events (SMQ) with COVID-19 Vaccine (ChAdOx1-S [recombinant]) COVID-19 Vaccine AstraZeneca (Other viral vaccines) EMA/PRAC/157045/2021.

https://www.ema.europa.eu/en/documents/prac-recommendation/signal-assessment-report-embolic-thrombotic-events-smq-covid-19-vaccine-chadox1-s-recombinant-covid_en.pdf

Line 107-108: the association of pre-existing immunity to Ad5 with the DIC isn’t convincing.

Done.

We changed the sentence and added the reference.

Reviewer 2 Report

The authors made all the necessary corrections. 

Author Response

Reviewer #2

We thank this Reviewer for her/his appreciating our paper

Reviewer 3 Report

Summary:

The manuscript titled “DIC-like Syndrome following Administration of ChAdOx1 nCov-19 Vaccination” is a case report on a patient developing DIC after receiving the first dose of the vaccine. The authors provide the case history of this patient including vitals, blood work analysis, and imaging results. The laboratory data indicated the patient had DIC and therefore was treated. The patient recovered and was released from hospitalization. In general the case report is detailed in the laboratory work. However it lacks in connecting this case to the broader population and the potential implications of adverse reactions from these adenovirus-based vaccine candidates against COIVD-19. Authors should expand on their discussion on the relevance of this study to COVID-19 vaccination efforts. I have the following comments:

Major comments:

The introduction section should be revised to provide a better flow of ideas and background information. The introduction begins with discussing the reports of blot clots associated with COVID vaccination, then discusses various COVID-19 vaccine platforms, then about the ChAdOx1 nCoV19 vaccine. The structure of this paragraph is not well organized and should be revised to better support the ideas the authors are trying to discuss.

The discussion section should also be revised. Again the flow of ideas and topics are not smooth for the reader to follow. Authors should further explain the reasoning to why they believe the patient developed DIC from vaccination.

Minor comments:

INTRODUCTION

Lines 30-33: The sentence is very long and clunky. Consider revising.

Lines 40-42: Please provide the reference for this sentence.

CASE HISTORY

Line 59: Remove the space between paragraphs.

Lines 61-62: Did the patient test negative for SARS-Cov-2? Please specify.

Line 65: Insert “and” after “…supra-aortic vessels,”

Line 70: Delete “e” after “…indicated in A, B”

Table 1: Please explain why December 13,2020 data is in the table. Did the patient get bloodwork done prior to the vaccination period?

Table 1: There is an asterisks after “international normalized ratio (0.9-1.2). Why is the asterisks for?

Line 86: The authors have previously defined the acronym DIC. You can remove “Disseminated intravascular coagulation”.

Line 87: Is the word “con” a typo?

Line 107: Did the authors mean increasing and not decreasing age?

Lines 112-114. Please elaborate on these two statements. The statements are good but need more explanation as to why the authors chose those biological markers to be reliable markers for the onset of DIC.

Author Response

Reviewer #3

We thank this Reviewer for the constructive comments and suggestions. Furthermore, we would like to really thank him/her for his/her appreciation about our research in the introduction section of his/her comments. This is our point to point reply.

Major comments:

The introduction section should be revised to provide a better flow of ideas and background information. The introduction begins with discussing the reports of blot clots associated with COVID vaccination, then discusses various COVID-19 vaccine platforms, then about the ChAdOx1 nCoV19 vaccine. The structure of this paragraph is not well organized and should be revised to better support the ideas the authors are trying to discuss.

Done.

The introduction section was re-structured and a table with the different vaccines available was added.

The discussion section should also be revised. Again the flow of ideas and topics are not smooth for the reader to follow. Authors should further explain the reasoning to why they believe the patient developed DIC from vaccination.

Done.

The discussion section was re-structured and 2 references were added.

Minor comments:

INTRODUCTION

Lines 30-33: The sentence is very long and clunky. Consider revising.

Done.

Lines 40-42: Please provide the reference for this sentence.

Done.

Line 59: Remove the space between paragraphs.

Done.

Lines 61-62: Did the patient test negative for SARS-Cov-2? Please specify.

Done.

Line 65: Insert “and” after “…supra-aortic vessels,”

Done.

Line 70: Delete “e” after “…indicated in A, B”

Done.

Table 1: Please explain why December 13,2020 data is in the table. Did the patient get bloodwork done prior to the vaccination period?

Laboratory tests closest to the event available are reported, in the absence of symptoms and before inoculation of the vaccine.

Table 1: There is an asterisks after “international normalized ratio (0.9-1.2). Why is the asterisks for?

The asterisk was removed.

Line 86: The authors have previously defined the acronym DIC. You can remove “Disseminated intravascular coagulation”.

Done.

Line 87: Is the word “con” a typo?

Typo corrected.

Line 107: Did the authors mean increasing and not decreasing age?

The sentence was changed.

Lines 112-114. Please elaborate on these two statements. The statements are good but need more explanation as to why the authors chose those biological markers to be reliable markers for the onset of DIC.

We consider the conclusion to be concise and direct.

Round 2

Reviewer 1 Report

the improvement is much limited, and the major concerns have not been responded at all. but the new problems are coming such as lines 107-108 of Binding of adenovirus to platelets, causing platelet activation and thrombosisand line 26 of to deliver mRNA SARS-CoV-2 Spike (S) protein. 

Reviewer 3 Report

No new comments/suggestions to add.

This manuscript is a resubmission of an earlier submission. The following is a list of the peer review reports and author responses from that submission.

Round 1

Reviewer 1 Report

The authors are presenting a case of DIC in a patient that received the first dose of AZ vaccine. It is of general interest, however the connections between DIC and vaccination might depend on various factors. Please change the title towards a more temperate tone.

General questions: was the patient tested for the presence of SARS-CoV-2?

Abstract is too long and it needs shortened. I believe some of the information can act easily as introduction. Also please do not use references in the abstract.

lines 48-59: please integrate into Table 1

lines 62-65: is pretty confusing what the authors are trying to say here, please rephrase

Lines 75-76: please include the antibody titers in Table 1

lines 91-95: without any references or evidence based on the study case, this is purely speculative.  Please rewrite this sentence.

Figure 1: please point the area(s) of interest and denominate each quadrant with A B C D

Reviewer 2 Report

It is beneficial to monitor and share the data about the potential risk related to the use of COVID-19 vaccines. Whatever the underling mechanism is clear or not, the introduction of the observed disease and the effective treatment is much informative to prevent or reduce hazardousness related to the vaccines. Therefore this case report about the disseminated intravascular coagulation (DIC) following the inoculation of ChAdOx1 nCoV-19, is of great importance. However, the data supporting the association between this vaccine and the reported DIC are not so compelling and the discussion on the likely reason is also away from the point. Therefore, it isn’t qualified to be pressed and needs to be further modified.

Major concerns:

  1. Besides the RBD-specific IgG response, it is invaluable to provide the data on the RBD-specific IgM and N-specific antibody responses too, and will further support the occurrence of DIC following the inoculation of the vaccine ChAdOx1 nCoV-19 rather than COVID-19.
  2. line 81-83: no data about thrombotic events are found in this mentioned reference 14.
  3. line 91-95: the hypothesis mentioned here as the explanation of the observed DIC isn’t reasonable and scientific and is pretty arbitrary. Even the patient did inoculate the vaccine, and the DIC ensued. To some extent, you can infer the vaccinated vaccine is responsible for the DIC, but how can you completely exclude the impact from the genetic differences. What is more, “In particular, the chimpanzee adenovirus could induce a procoagulative state and a subsequent DIC due to the lack of previous exposure to the specific adenovirus”, what do you depend logically and experimentally to establish the association between the DIC and exposure to the ChAd. If it is true, why all the remainder, same free from the infection of the ChAd as the patient, don’t develop DIC after the vaccination of ChAdOx1 nCoV-19.
  4. line 105-106: It is not likely the low pre-existing immunity to the vaccine vector Ad5 virus works in this case, the adenoviral vector used here is different from Ad5, unless the author can provide related information and materials to support this point.

Minor concerns:

  1. Please pay attention to the difference and reasonable usage between and of the virus of SARS-CoV-2 and the resultant disease of COVID-19. For example, the expressions of anti-COVID-19 vaccine and anti-SARS-CoV2 vaccine aren’t suitable academically and in writing.
  2. The values of D-dimer, tested during her hospitalization and listed in Table1, should be translated into ng/ml same as the reference range.
  3. line 80: 1.000 should be 1,000.